# Impact of Corticosteroid Administration within 7 Days of the Hospitalization for Influenza Pneumonia with Respiratory Failure: A Propensity Score Analysis Using a Nationwide Administrative Database

**DOI:** 10.3390/jcm10030494

**Published:** 2021-01-31

**Authors:** Daisuke Okuno, Takashi Kido, Keiji Muramatsu, Kei Tokutsu, Sakiko Moriyama, Takuto Miyamura, Atsuko Hara, Hiroshi Ishimoto, Hiroyuki Yamaguchi, Taiga Miyazaki, Noriho Sakamoto, Yasushi Obase, Yuji Ishimatsu, Yoshihisa Fujino, Kazuhiro Yatera, Shinya Matsuda, Hiroshi Mukae

**Affiliations:** 1Department of Respiratory Medicine, Nagasaki University Graduate School of Biomedical Sciences, Nagasaki 852-8501, Japan; vkvkv10101@gmail.com (D.O.); sakiko.tanaka.0@gmail.com (S.M.); t-miyamura@nagasaki-u.ac.jp (T.M.); clover0409ah@gmail.com (A.H.); h-ishimoto@nagasaki-u.ac.jp (H.I.); hyamaguchi@nagasaki-u.ac.jp (H.Y.); nsakamot@nagasaki-u.ac.jp (N.S.); obaseya@nagasaki-u.ac.jp (Y.O.); hmukae@nagasaki-u.ac.jp (H.M.); 2Department of Preventive Medicine and Community Health, University of Occupational and Environmental Health, Japan, Kitakyushu 807-8555, Japan; km@med.uoeh-u.ac.jp (K.M.); keitokutsu@med.uoeh-u.ac.jp (K.T.); smatsuda@med.uoeh-u.ac.jp (S.M.); 3Department of Infectious Diseases, Nagasaki University Graduate School of Biomedical Sciences, Nagasaki 852-8501, Japan; taiga-m@nagasaki-u.ac.jp; 4Department of Nursing, Nagasaki University Graduate School of Biomedical Sciences, Nagasaki 852-8520, Japan; yuji-i@nagasaki-u.ac.jp; 5Department of Environmental Epidemiology, Institute of Industrial Ecological Science, University of Occupational and Environmental Health, Japan, Kitakyushu 807-8555, Japan; zenq@med.uoeh-u.ac.jp; 6Department of Respiratory Medicine, University of Occupational and Environmental Health, Japan, Kitakyushu 807-8555, Japan; yatera@med.uoeh-u.ac.jp

**Keywords:** ARDS, infection and inflammation, influenza, viral infection

## Abstract

Influenza pneumonia, which causes acute respiratory distress syndrome and multiple organ failure, has no established management protocol. Recently, corticosteroid therapy was used to treat coronavirus disease 2019 with respiratory failure; however, its effectiveness as a treatment for influenza pneumonia remains controversial. To investigate the impact of corticosteroid therapy for the early phase of severe influenza pneumonia, we compared influenza pneumonia patients with respiratory failure treated with or without corticosteroids within 7 days after hospital admission using a Japanese nationwide administrative database. The primary endpoint was the mortality rate. The secondary endpoints were duration of intensive-care unit management, invasive mechanical ventilation, and hospital stay. The inverse probability weighting method with estimated propensity scores was used to minimize the data collection bias. We included 3519 patients with influenza pneumonia with respiratory failure. Of these, 875 were treated with corticosteroids. There was no significant difference between the groups regarding 30-day and 90-day mortality, duration of intensive-care unit management, invasive mechanical ventilation, and hospital stay. However, the in-hospital mortality rate was higher in the corticosteroid group. The use of systematic corticosteroid therapy in patients with influenza pneumonia was associated with a higher in-hospital mortality rate.

## 1. Introduction

The influenza virus is a common pathogen causing upper respiratory tract symptoms, high fever, chills, and myalgia. Occasionally, it can cause severe disease with reported fatality rates ranging from 17.4% to 46% in different countries during the A/H1N1 2009 pandemic [1]. Dysregulated systemic inflammation is associated with disease severity and can cause acute respiratory distress syndrome (ARDS) and/or multiple organ failure, leading to a poor prognosis [2,3,4,5,6,7].

The management of severe influenza has not been established. Systemic corticosteroid therapy could inhibit excessive inflammatory cytokine production and improve critical illness prognosis, even in infectious diseases [8]. The clinical efficacy of systemic corticosteroid therapy was shown in patients with community-acquired pneumonia (CAP) [9,10,11,12,13,14] and coronavirus disease 2019 (COVID-19) with respiratory failure [15], renewing the attention to its efficacy in severe viral pneumonia. For patients with severe influenza, retrospective and observational studies have found no benefit for corticosteroid therapy when assessing the mortality rate [16,17,18,19,20,21], with 39.9% to 70.8% of patients treated with corticosteroids [19,20,21,22,23,24,25].

Here, we investigated evidence from clinical practice on the use and effectiveness of corticosteroid therapy within 7 days from admission for influenza pneumonia patients with respiratory failure, using the Diagnostic Procedure Combination (DPC), a Japanese nationwide administrative database. We employed the inverse probability weighting (IPW) method with estimated propensity scores to minimize the bias introduced by baseline characteristics of retrospectively collected data on the corticosteroid and non-corticosteroid groups [26,27,28,29].

## 2. Experimental Section

### 2.1. Data Source

We used the DPC database to investigate the effectiveness of corticosteroid treatment for influenza pneumonia with respiratory failure. All data in the DPC database were gathered by the DPC research group, which receives funding from the Japanese Ministry of Health, Labour, and Welfare. The DPC is a Japanese case-mix patient classification system launched in 2002 for payment management and modernization of the healthcare system [30]. It covered approximately 80% of all acute care inpatient hospitalizations in Japan in 2019 and was used to evaluate treatment effects and complications [31,32,33,34,35,36,37]. The database contains the following details: patient age, sex, diagnosis, comorbidities at admission and during hospitalization (coded following the International Classification of Diseases, 10th revision; ICD-10), state of consciousness according to the Japan Coma Scale (JCS), medical procedures, medications, intensive-care unit (ICU) admission, interventional procedures (including hemodialysis, mechanical ventilation, and heart-lung medicine), length of hospital stay, and discharge status (including in-hospital deaths) [30,33]. The requirement for informed consent was waived because any personally identifiable information was removed from the extracted data and because of the retrospective nature of the study. This study was approved by Ethics Committee of Medical Research, University of Occupational and Environmental Health, Japan and conducted according to guidelines in the Declaration of Helsinki.

### 2.2. Patient Selection

Influenza pneumonia with respiratory failure was defined as influenza pneumonia (ICD-10 code J10.0 or J11.0) or influenza (ICD-10 code J10.1 or J10.8) with ARDS (J80), and using either oxygen therapy, nasal high flow oxygen therapy (NHF), non-invasive positive pressure ventilation (NPPV), invasive mechanical ventilation (IMV), or extracorporeal membrane oxygenation (ECMO) within 7 days from admission. The data of patients admitted between April 2016 and March 2018 who met the criteria were extracted from the DPC database. The data were divided into the corticosteroid and non-corticosteroid groups. The patients who received intravenous (IV) or oral corticosteroids (betamethasone sodium phosphate, cortisone acetate, dexamethasone sodium phosphate, fludrocortisone acetate, hydrocortisone sodium succinate, methylprednisolone, prednisolone, or triamcinolone) within 7 days from admission were included in the corticosteroid group.

### 2.3. Variables

The following factors were used as variables: age (in years), sex, advanced treatment hospital (hospitals required to provide advance medical care, most of which are university hospitals), emergency admission, emergency transport, home healthcare, hospital volume (number of patients who met the inclusion criteria between April 2016 and March 2018, pregnancy, smoking, diagnosis of asthma, cancer, cardiovascular disease (diagnosed as acute myocardial infarction, cardiac valvular disease, cardiomyopathy, or pulmonary embolism), cerebrovascular disease (diagnosed as cerebral hemorrhage subarachnoid hemorrhage, or cerebral infarction), chronic kidney disease, chronic respiratory failure, chronic obstructive pulmonary disease (COPD), diabetes mellitus, liver dysfunction (diagnosed as hepatitis, liver cirrhosis, or liver failure), neurological dysfunction (JCS score at admission of ≥100 (indicating a response by closing eyes, no verbal response, and movement in response to pain [33]. Though the DPC data does not include the Glasgow Coma Scale (GCS), a JCS score of 100 is approximately equal to GCS E1V1M5)), heart failure, hypertension, interstitial lung diseases, medication use (albumin, antithrombin III, heparin, immunoglobulin, insulin, transfusion of platelets or red blood cells, recombinant human soluble thrombomodulin (rhTM), sivelestat, vasopressors, oseltamivir, zanamivir, laninamivir, peramivir, azithromycin (IV or oral), clarithromycin, erythromycin (IV or oral), carbapenem (IV), cephalosporin (1st–4th generation, IV), clindamycin (IV), minocycline (IV), metronidazole (oral), new quinolone (IV or oral), penicillin antibiotics (IV), anti-MRSA drug (IV)), emergency hemodialysis or maintenance hemodialysis, ECMO, number of IMV days, NPPV, oxygen therapy, use of polymyxin B-immobilized fiber column (PMX), and ICU admission within 7 days from hospital admission.

### 2.4. OUTCOMES

The primary endpoint was set as mortality (in-hospital mortality, 30 day-mortality, and 90 day-mortality). The secondary endpoints were the durations through the entire hospitalization period of ICU management, mechanical ventilation, and hospital stay.

### 2.5. STATISTICAL ANALYSIS

The propensity score was calculated by a logistic model with baseline variables that could affect the administration of corticosteroids, including all variables described above. The C-statistic (area under the operating characteristic curve) was employed to evaluate the goodness of fit. These methods were conducted using IBM SPSS 22.0 (IBM Corp., Armonk, NY, USA). Then, we adjusted the covariates and evaluated the outcomes between the corticosteroid and non-corticosteroid groups by the inverse probability weighting method [33,36,38]. The covariates before adjustment were evaluated using the chi-squared (χ^2^) test for categorical variables and the unpaired *t*-test for continuous variables. These methods were conducted using STATA/IC 14.0 (StataCorp, College Station, TX, USA). Differences with *p* < 0.05 were considered statistically significant in all tests.

## 3. Results

A flow chart of influenza pneumonia patient selection is shown in Figure 1. The data of 5357 patients with influenza pneumonia or influenza with ARDS were extracted from the DPC database as described in the Methods. Of these, 1838 patients without respiratory failure were excluded, while 3519 patients were found eligible for this study. Of these 3519 patients, 875 (24.8%) were treated with corticosteroids within 7 days from admission, and 2644 (75.2%) were not. In the corticosteroid group, 86 patients received high-dose corticosteroid treatment (intravenous administration of methylprednisolone at a dose of >500 mg/day for >1 day), 679 received other intravenous corticosteroids, and 288 received oral corticosteroids. There was partial overlap among these.

Subsequently, we used the IPW method with estimated propensity scores, and the C-statistic of the propensity score was 0.787. The baseline characteristics of patients before and after they were adjusted for confounders are presented in Table 1. Before adjustment, the baseline variables of age, advanced treatment hospital, emergency admission, home healthcare, asthma, cerebrovascular disease, chronic respiratory failure, COPD, liver dysfunction, heart failure, hypertension, interstitial pneumonia, albumin, antithrombin III, heparin, immunoglobulin, insulin, platelet transfusion, red cell transfusion, rhTM, sivelestat administration, vasopressor, zanamivir, peramivir, azithromycin (IV), clarithromycin, carbapenem, new quinolone (IV), IMV days, NPPV, oxygen therapy, PMX, and ICU management differed significantly between the groups. After adjusting for confounders, the patient baseline characteristics in the two groups were similar across these variables, except for liver dysfunction, which remained significantly different.

The results of the primary and secondary endpoints of patients treated with or without corticosteroids as assessed by the IPW method with the estimated propensity score are shown in Table 2. Although there was no difference in 30-day mortality (14.52% vs. 13.38%, *p* = 0.454) and 90-day mortality (18.43% vs. 15.08%, *p* = 0.054), the in-hospital mortality rate was significantly higher in the corticosteroid group (19.21% vs. 15.36%, *p* = 0.031). There was no difference between the groups throughout hospitalization in durations of ICU management (0.60 ± 0.079 vs. 0.58 ± 0.065 days, *p* = 0.857), IMV (2.13 ± 0.23 vs. 2.39 ± 0.23 days, *p* = 0.373), and hospital stay (19.90 ± 1.07 vs. 19.38 ± 0.44 days, *p* = 0.649). To account for multiple testing, we performed the Hochberg’s test with a false discovery rate of 0.10 [39], and the results were unchanged.

## 4. Discussion

The management of severe influenza pneumonia with or without ARDS has not been established, and the efficacy of early corticosteroid therapy remains unclear. Following the recent favorable outcome of corticosteroid therapy for COVID-19 with respiratory failure, much attention has been given to its efficacy when treating severe viral pneumonia. In the present study, we investigated clinical practice evidence of the use and effectiveness of corticosteroid therapy in a study of 3519 influenza pneumonia patients with respiratory failure using the Japanese nationwide administrative database. Surprisingly, a quarter of the patients (875/3519, 24.8%) received systemic corticosteroid therapy within 7 days from hospital admission. Previous studies have reported that adjuvant corticosteroid therapy was used in 39.9% to 69.0% of patients with severe influenza pneumonia requiring ICU management or mechanical ventilation [19,20,21,24,25]. In this study, although we included relatively mild cases that only required oxygen therapy, 24.8% of the patients were treated with corticosteroids, suggesting that corticosteroid therapy is commonly used for influenza pneumonia with respiratory failure. 

However, the rate of in-hospital mortality was significantly higher in the corticosteroid group (19.21% vs. 15.36%, *p* = 0.031) after adjusting for confounders by the IPW method, using the estimated propensity score. In a previous study on pH1N1 pneumonia with acute respiratory failure that required admission to the ICU, no differences in mortality were noted between patients who received corticosteroids and those who did not (18.38% vs. 17.37%, *p* = 0.806) [17]. Corticosteroid therapy was also associated with a higher mortality rate in patients with pH1N1 infection admitted to the ICU (54% vs. 31%, *p* = 0.004) [20] and in patients with severe respiratory failure requiring mechanical ventilation (33.7% vs. 16.8%, *p* = 0.004) [21]. A meta-analysis also showed that corticosteroid therapy was related to a higher mortality rate (odds ratio 1.98, 95% CI: 1.62–2.43; *p* < 0.001) [1]. In our results, corticosteroid therapy did not affect the duration of ICU management, IMV, or hospital stay. It was shown that corticosteroid therapy had unfavorable effects in patients with pH1N1-related critical illness requiring ICU admission. The ICU management duration was 13.5 vs. 8.8 days, mechanical ventilation was 13.3 vs. 9.6 days, and hospital stay was 30.8 vs. 18.4 days [20]. In a meta-analysis, corticosteroid treatment was associated with longer ICU management (weighted mean differences 4.78, 95% CI: 2.27–7.29, *p* < 0.001) and mechanical ventilation (weighted mean differences 3.82, 95% CI: 1.49–6.15, *p* = 0.001) [1]. Collectively, our results and those of previous studies suggest that corticosteroid therapy does not have favorable effects on the mortality rate, duration of ICU management, IMV days, or the length of hospital stay. 

Thirty-day mortality was not different, and 90-day mortality was higher in the corticosteroid group, although there was no statistically significant difference. The in-hospital mortality was significantly higher in the corticosteroid group. These results suggest that corticosteroids have a negative effect on a longer mortality period, because the in-hospital mortality has a longer prognosis than 30-day and 90-day mortality. This might be explained by complications arising from long-term corticosteroid therapy, including hospital-acquired bacterial pneumonia or invasive fungal infections [19,20,21,40]. Corticosteroid therapy influences immune cells such as macrophages and granulocytes that protect the host from bacteria, thus increasing the incidence of infections [41]. Kim et al. reported that patients receiving corticosteroid therapy for influenza pneumonia more frequently developed secondary bacterial pneumonia (57% vs. 22%) and invasive pulmonary aspergillosis (4% vs. 0%) than patients receiving non-corticosteroid therapy [20]. Martin-Loeches et al. and Brun-Buisson et al. also demonstrated that corticosteroid therapy was associated with a higher incidence of pneumonia than non-corticosteroid therapy (26.2% vs. 13.8% and 41.0% vs. 26.4%, respectively) [19,21]. Furthermore, *Pseudomonas aeruginosa* and *Acinetobacter baumannii* were often identified as the pathogens causing hospital-acquired pneumonia in patients treated with corticosteroids for severe influenza pneumonia [19], suggesting that corticosteroid therapy increases the risk of contracting an opportunistic and refractory infection. Some experimental data on influenza and severe acute respiratory syndrome-related coronavirus have also shown that corticosteroid therapy might prolong viral clearance and create a suitable environment for expanded viral replication, subsequently leading to a higher plasma viral load [16,42,43,44].

This study has limitations similar to those of previous studies using the DPC database [31,32,33,45]. First, it was an observational retrospective study without randomization. Although adjusting the baseline characteristic differences using the IPW method with the estimated propensity score reduced the effect of this limitation, there might still be confounders that we were unable to evaluate with the DPC database and could have influenced the outcome. Examples of such confounders are vital signs, laboratory findings, radiological findings, and mechanical ventilation settings. Second, we showed that 86 patients received high-dose corticosteroid treatment, 679 patients received other intravenous corticosteroids, and 288 patients received oral corticosteroids. More detailed information on the dosage of corticosteroids is difficult to estimate because of its complexity. Third, the influence of secondary bacterial pneumonia is unknown because microbial etiology is not included in the DPC database. Fourth, we cannot exclude the influence of the different virus types on the clinical outcome, as this information is not included in the DPC database. However, information regarding the popular influenza virus type during each season in Japan is available in the database of the Japanese National Institute of Infectious Diseases (https://www.niid.go.jp/niid/en/influenza-e.html). According to the data presented in this database, the A/H1, A/H3, and B rates were 4%, 78%, and 18% in years 2016/2017 (September 2016 to August 2017), and 23%, 32%, and 45% in years 2017/2018 (September 2017 to August 2018), respectively. Thus, we believe that A/H1, A/H3, and B were the major virus types present in our study (data analyzed from April 2016 to March 2018). Despite these limitations, this study has the major advantage of using a large dataset for corticosteroid therapy efficacy evaluation.

## 5. Conclusions

We provided clinical practice evidence on the usage and effectiveness of corticosteroid therapy by analyzing the data of 3519 influenza pneumonia patients with respiratory failure with using a Japanese nationwide administrative database. A quarter of the patients (875/3519, 24.8%) received corticosteroid therapy within 7 days from admission, which was associated with a higher in-hospital mortality rate. Although this observational retrospective study has some limitations, it has benefited from using a large dataset to investigate the actual use and effects of corticosteroid therapy in patients with influenza pneumonia with respiratory failure.

## Figures and Tables

**Figure 1 jcm-10-00494-f001:**
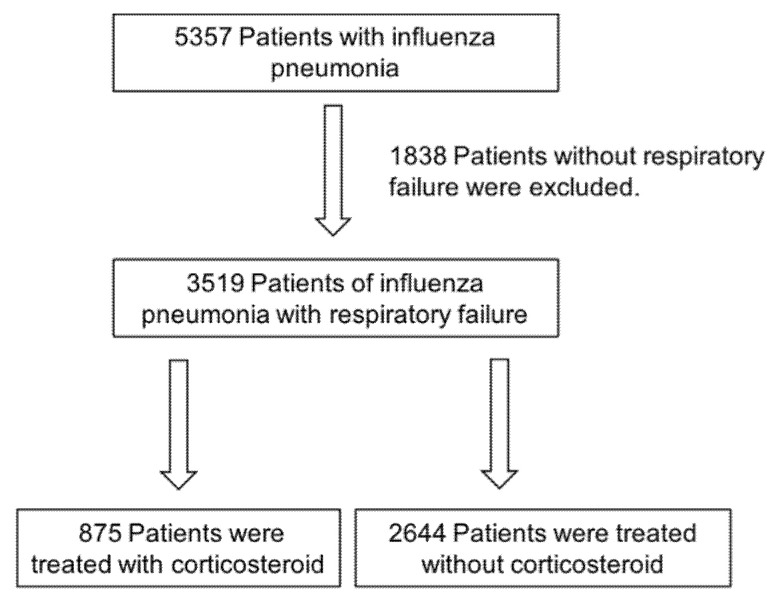
Flow chart of patient selection.

**Table 1 jcm-10-00494-t001:** Baseline characteristics of patients treated with or without corticosteroids before and after adjusting for confounders using the propensity score weighting method.

Baseline Characteristics	Before Adjustment			After Adjustment		
Variables	Corticosteroid *n* = 875	Non-Corticosteroid *n* = 2644	*p*-Value	Corticosteroid *n* = 875	Non-Corticosteroid *n* = 2644	*p*-Value
Age, years	62.9 ± 1.0	71.8 ± 0.5	<0.001	68.4 ± 0.7	69.3 ± 0.5	0.068
Sex (Female)	41.4	44.4	0.113	45.4	43.6	0.136
Advanced treatment hospital	10.4	7.5	0.006	8.0	8.4	0.421
Emergency admission	74.9	70.5	0.012	73.2	71.6	0.130
Emergency transport	55.5	59.2	0.055	55.9	57.3	0.200
Home healthcare	6.6	10.9	<0.001	9.3	9.7	0.599
Hospital volume	12.9 ± 0.7	14.1 ± 0.5	0.110	12.9 ± 0.6	13.6 ± 0.4	0.110
Pregnancy	0	0.04	0.565	0	0.03	0.317
Smoking	37.5	34.9	0.174	35.2	36.3	0.284
Asthma	27.2	8.2	<0.001	13.9	14.0	0.851
Cancer	8.6	8.1	0.682	8.6	8.4	0.766
Cardiovascular disease	3.5	5.0	0.084	4.4	4.9	0.459
Cerebrovascular disease	6.4	11.2	<0.001	9.8	9.6	0.812
Chronic kidney disease	9.8	9.4	0.695	9.2	9.5	0.640
Chronic respiratory failure	3.2	2.0	0.034	2.7	2.2	0.095
COPD	12.2	5.8	<0.001	8.2	8.2	0.992
Diabetes mellitus	18.4	17.1	0.378	19.1	18.4	0.547
Liver dysfunction	29.7	16.3	0.013	22.4	18.4	0.045
Neurological dysfunction	6.7	5.5	0.167	5.7	6.1	0.553
Heart failure	12.9	17.4	0.002	15.9	15.5	0.624
Hypertension	19.8	25.7	<0.001	23.8	24.3	0.686
Interstitial lung disease	18.7	11.3	<0.001	13.0	13.2	0.666
Albumin	10.1	3.1	0.001	5.1	5.5	0.467
Antithrombin III	3.4	0.6	<0.001	1.6	1.8	0.624
Heparin	21.1	10.6	<0.001	13.0	13.7	0.276
Immunoglobulin	4.7	1.5	<0.001	3.3	2.9	0.550
Insulin	31.2	12.8	<0.001	18.0	18.1	0.877
Platelet transfusion	4.0	1.1	<0.001	2.4	2.7	0.625
Red cell transfusion	6.1	2.8	<0.001	4.5	4.2	0.575
rhTM	5.9	1.2	<0.001	2.7	2.6	0.796
Sivelestat	1.7	0.5	0.001	1.0	0.9	0.514
Vasopressor	23.3	10.7	<0.001	14.0	14.3	0.565
Oseltamivir	10.2	9.7	0.698	10.7	10.0	0.380
Zanamivir	0.7	0.1	0.001	0.3	0.5	0.378
Laninamivir	1.6	2.5	0.110	2.6	2.2	0.388
Peramivir	67.1	60.6	0.001	61.6	62.3	0.550
Azithromycin (iv)	9.3	3.6	<0.001	4.6	4.6	0.920
Azithromycin (oral)	0.1	0.4	0.119	0.4	0.4	0.549
Clarithromycin	5.2	3.7	0.045	4.2	4.2	0.908
Erythromycin (oral)	0.9	0.4	0.081	0.7	0.8	0.601
Erythromycin (iv)	0.1	0.04	0.411	0.05	0.05	0.854
Carbapenem (iv)	19.5	10.4	<0.001	13.0	13.4	0.481
Cephalosporin (1st generation, iv)	0.9	1.3	0.380	1.6	1.3	0.578
Cephalosporin (2ndgeneration, iv)	1.4	1.6	0.598	1.1	1.5	0.114
Cephalosporin (3rdgeneration, iv)	32.2	32.2	0.998	33.0	32.0	0.337
Cephalosporin (4thgeneration, iv)	1.8	1.5	0.518	1.7	1.8	0.680
Clindamycin (iv)	0.7	0.6	0.693	0.8	0.5	0.400
Minocycline (iv)	2.5	1.9	0.228	2.5	2.1	0.293
Metronidazole (oral)	0.3	0.2	0.556	0.3	0.2	0.401
New quinolone (iv)	12.9	6.0	<0.001	9.1	8.6	0.466
New quinolone (oral)	5.9	5.2	0.387	5.4	5.7	0.580
Penicillin antibiotics	35.9	37.3	0.467	37.1	37.0	0.891
anti-MRSA drug (iv)	3.9	1.2	<0.001	2.0	2.0	0.929
Emergency hemodialysis	0.6	1.0	0.258	0.7	0.9	0.377
Maintenance hemodialysis	14.9	25.7	0.063	1.8	2.2	0.177
ECMO	0	0.1	0.319	0	0.08	0.083
IMV wearing days	1.5 ± 0.1	0.5 ± 0.0	<0.001	0.8 ± 0.0	0.8 ± 0.0	0.754
NPPV	1.5	0.6	0.008	0.9	0.8	0.391
Oxygen therapy	82.2	92.9	<0.001	90.4	90.1	0.526
PMX	0.3	0.04	0.020	0.1	0.09	0.352
ICU admission rate	11.5	4.6	<0.001	6.9	7.3	0.499

Data are presented as the % or mean ± (standard error). Groups were adjusted using the propensity score weighting method. Abbreviations: COPD, chronic obstructive pulmonary disease; ECMO, extracorporeal membrane oxygenation; ICU, intensive-care unit; IMV, invasive mechanical ventilation; iv, intravenous administration; JCS, Japan Coma Scale; NHF, nasal high flow oxygen therapy; NPPV, non-invasive positive pressure ventilation; oral, oral administration; PMX, polymyxin B-immobilized fiber column; rhTM, recombinant human soluble thrombomodulin.

**Table 2 jcm-10-00494-t002:** Results of the primary and secondary endpoints of patients treated with or without corticosteroids.

Outcomes	Corticosteroid	Non-Corticosteroid	*p*-Value
30-day mortality (%)	14.52	13.38	0.454
90-day mortality (%)	18.43	15.08	0.054
In-hospital mortality (%)	19.21	15.36	0.031
ICU management, mean days (SE)	0.60 (0.08)	0.58 (0.07)	0.857
IMV, mean days (SE)	2.13 (0.23)	2.39 (0.23)	0.373
Hospital stay, mean days (SE)	19.90 (1.07)	19.38 (0.44)	0.649

Data are presented as the % or mean (standard error). Groups were adjusted using the propensity score weighting method. Abbreviations: ICU, intensive-care unit; IMV, invasive mechanical ventilation; SE, standard error.

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
