# Peer review of "Impact of Corticosteroid Administration within 7 Days of the Hospitalization for Influenza Pneumonia with Respiratory Failure: A Propensity Score Analysis Using a Nationwide Administrative Database"

_jcm, 2021, doi:10.3390/jcm10030494_

Round 1
Reviewer 1 Report
Daisuke Okuno et al. examined the impact of corticosteroid therapy in severe influenza pneumonia by using a propensity score from a Japan nationwide database. Influenza pneumonia is a timely major topic that deserves a lot of attention. This manuscript has some strengths in particular a nationwide database that included a total of 3,519 patients. However there some important findings that need to be corrected as it reads below:
- The manuscript title misguides. It reads early corticosteroid in severe pneumonia. It's difficult to agree with the statement of "early corticosteroid" that it was given within 7 days of the hospitalization. Second, severe influenza pneumonia is not supported by the study data given the lack of use of assessments/scores of severity such as APACHE or SOFA or qSOFA for ICU, and CURB-65 for CAP
- Variables sections, line 92, reads ARDS as one of the variables. However, ARDS data is not found in the results.
- Statistical analysis, line 116, "we evaluated the the average treatment effect of covariates and outcomes" Unfortunately, this is not clear.
- It's not clear the number of patients admitted in the ICU. Table 1 reads ICU admission 11.5 that it's not clear what was the measure unit: ? days, %, etc.
- ARDS should be one of the variables in table 1.
- Discussion, line 196 "but affects the longer-term prognosis (in-hospital mortality" This is unclear.
- Conclusions, line 229: ..."3519 patients with severe influenza.." This is not correct.
Unfortunately, this manuscript has too many findings that it can't be accepted for publication.
Reviewer 2 Report
This Article by Daisuke Okuno et al. describe an important analysis of the impact of early corticosteroid therapy in sever influenza pneumonia.
This article suffers from limitations, partly discussed in discussion part of the manuscript, but some major limitations remain.
- How the inclusion period been determined by the authors? Why the authors considered one complete and two partly complete influenza season ? Was the number of patient needed to include determined before the analysis?
As the study include patients from two successive year, the "hospital volume" should take into account the data for these two years of be splitted into two different parts.
Why have the authors considered JCS score for neurological dysfunction and not consensual criteria as the Glasgow coma scale?
One major limitation of the study is that the results were not stratified according to the influenza virus type or influenza season. It was clearly demonstrated that different virus types lead to different clinical outcome (and response to treatment). Authors have to complete it.
How the authors consider to take into account the multiplication of the tests they used? Have they considered a multiple testing correction, that risk to limit significant results?
How the authors decided to split the categories? For example, it could be interesting to consider antibiotic use or no as a major category before to split the data per molecules. Similar, split molecules categories depending on the way of administration could be considered as excessive splitting.
Absence of effect of Corticosteroids on 90-day mortality seems to be due to a lack of power, as there seem to be a tendency. authors have to discuss it.
Please indicate "et al." in italic.
Line 224 : the statement "this limitation was compensated by evaluating the administrated antibiotics" is clearly false and have to be modified and/or discussed, as this limitation was clearly not been completely compensated.
Round 2
Reviewer 2 Report
Even if I do not completely agree with the response to the comment #5, I think that the manuscript is know suitable for publication after having taken into account the following comment. I think that the manuscript would benefit of an analysis of baseline characteristics pooling iv and oral administration of molecules, as, even if I agree with the authors' response, administration is not only a question of severity. Moreover, I think that this pooling would allow an higher power for this comparison, and their results would be interesting.
